# Earth-Bound Preaching: Engaging Scripture, Context, and Indigenous Wisdom

HyeRan Kim-Cragg

Emmanuel College, 75 Queen's Park Cres, Toronto, ON M5S 1K7, Canada; hyeran.kimcragg@utoronto.ca

**Abstract:** In developing an Earth-bound homiletics, three homiletical movements are suggested: engaging Scripture, engaging global and local situatedness, and engaging the Indigenous worldview of "all my relations" by tapping into Indigenous knowledge. These three movements need not take place in any chronological order, nor should they be seen as a hierarchy. Rather, they are complementary and interconnected. The author, before articulating these movements, offers reasons for why the topic of the climate crisis is not preached on and then addresses the challenge of selecting biblical texts, delineating the strengths and weaknesses of using the lectionary readings versus a preacher's individual choices. The article further addresses the danger of biblical literalists who deny global warming. Each homiletical movement will be elaborated using actual sermons as concrete examples of Earth-bound homiletics.

**Keywords:** agency of the creation; all my relations; climate crisis; Earth-bound homiletics; indigenous epistemology; lectionary; lifeways

## 1. Introduction

This article is written on the premise that our planet Earth is in the midst of a climate crisis and that preachers have the responsibility to exhort congregations to be attentive to this crisis and to act. Using a practical theological method, namely, paying attention to the particular situatedness of the people of faith, the author proposes an "Earth-bound" homiletics encompassing a circular homiletical movement inspired by the very nature of the Earth, which spins as it orbits the sun. The most effective way to tackle climate change and the environmental crisis is not linear but cyclical. Before delving into these movements, first, the author presents some reasons for why the climate crisis has not been a major topic in Christian theological disciplines and is rarely preached upon in most congregations of any denomination. Then, the problem of choosing Scripture readings for Earth-bound preaching is posed. Turning to three homiletical movements, each movement is articulated with sermon examples employing a practical theological case study method (Schipani 2011, pp. 91–101). Movement One engages the hermeneutical reading of the text from a creation-centered stance as an essential and integral part of homiletical exegesis. Movement Two attempts an interdisciplinary engagement underscoring the situatedness of the audience with attention to the "glocal" situatedness in both "global" and "local" contexts. Movement Three seeks to tap into Indigenous knowledge alluded to in the phrase "all my relations", meaning knowledge that honors the agency of the creation and its relational nature. The author proposes that this creation–human interconnected epistemology can be incorporated not only in the content but also in the form of sermon making. In sum, Earth-bound homiletics aims to proclaim hope in relational ways, noting that the Good News is for all life on Earth!

## 2. Problem of Choosing Scripture for Earth-Bound Preaching

The climate crisis is one of the most pressing issues today that is affecting both humans and their other-than-human kin in unprecedented ways. The impact of this crisis is huge,

yet it is not fully and faithfully addressed in preaching, especially in the North Atlantic regions (Jones et al. 2014, p. 4). There is little in the homiletical literature dealing with the climate crisis (Hessel 1985; Achtemeier 1992; LeQuire 1996; Rhoads 2007; Habel et al. 2011; Holbert 2012; Schade 2015). The Academy of Homiletics, a guild that has been established and active since 1965, did not have a workgroup focused on the climate crisis until 2022. This lack of scholarly attention is not a problem only in the field of homiletics. Practical theology, the larger discipline to which homiletics belongs, has not responded to the issue, either (McCarroll and Kim-Cragg 2023). The most recently published encyclopedic handbook of practical theology omits ecological issues, including the climate crisis, as a subject. Even though it deals with close to sixty topics in over 800 pages, the editors admit that there are a number of "blind spots and blanks" (Weyel et al. 2022, p. 1). While an eco-feminist perspective is noted in one chapter that focuses on preaching, it is not given nearly the attention it deserves. This is despite the fact that, as Sunggu Yang says, "For the foreseeable future . . . eco-feminist concerns will remain one of the most critical issues that the Christian pulpit should address as we now face unprecedented environment disasters and many old and new forms of misogynist oppressions" (Yang 2022, p. 452).

Looking at the weekly preaching life of Christian congregations, the reality is not much better. The climate crisis and ecological concerns only occasionally arise. With the exception of two Sundays per year, the week of Earth Day in April and the Sunday remembering St. Francis in October, few preachers take the opportunity to preach with ecological accents, so to speak. This constitutes a less-than-adequate treatment of this urgent global problem. The burden of responsibility should not be solely on the shoulders of the preachers or worship leadership teams in local congregations, however. Failure to preach on the climate crisis in substantial and serious ways may lead to the reduction of broader public engagements with those who are outside faith communities about climate or other environmental issues.

One of the most serious systemic problems in preaching is related to selecting Scripture. Roman Catholic, Orthodox, Anglican, Lutheran, and most mainline Protestant denominations follow a lectionary which was not created with a mind to address the climate crisis or ecological issues. This lack of intention to address the climate crisis should not be regarded as a dismissal of the strengths and the contributions of the lectionary as a way to inform the selection of Scripture for preaching, as there are many strengths in using a lectionary for preaching (Kim-Cragg 2023a, p. 34). The creation of various lectionaries as a selection of readings of Scripture in order to guide their use in worship is an ancient practice that derives from Jewish traditions. The vast majority of mainline Protestant churches use the Revised Common Lectionary (RCL 1992) in the United States and Canada today. It emerged out of an ecumenical effort based on the Roman Catholic Lectionary for Mass (1971), which was created following the reforms of the Second Vatican Council (Burns and Kim-Cragg 2023, pp. xxvi–xxxvi). Its ecumenical collaborative and comprehensively positive effects notwithstanding, including producing useful lectionary-based commentaries[1], relying on the Revised Common Lectionary as the only way to select the Bible passages for preaching is limiting as far as addressing the climate crisis is concerned.

In the evangelical and Pentecostal traditions, most preachers do not follow a set of lectionary readings but afford themselves the freedom to choose their own passages for preaching. The free selection of texts has strengths, as it can underscore local contexts and may well serve individual congregations' needs better than a strict adherence to a lectionary. However, there are potential problems as well. For one, preachers may neglect larger parts of the canon and Christian doctrine when they rely on their favorite verses and turn repeatedly to the same text. Preachers and worship leaders tend to select texts that make the congregation feel good and avoid texts that are ethically challenging. "Consequently," Ronald Allen writes, "free selection and local plan often have the effect of reinforcing the theological, intellectual, moral, and institutional status quo" (Allen 1998a, p. 113). The status quo may include the failure to address the climate crisis. An ecological vision may be constricted if the preacher is not challenged to move outside their comfort zone (Bouma-Prediger 2010).

Tanya Riches and Anna Kirkpatrick-Jung have studied the limitations of the free selection of Scripture texts not just for preaching but also for worship broadly when it comes to the ecological crisis and megachurches. When it comes to megachurches, they note that intercessory prayer is rarely, if ever, directed to ecological issues, and this omission is obvious particularly in Pentecostal spirituality. From prayer to preaching, the Earth is missing in worship. However, they argue that the Pentecostal tradition, a tradition that emphasizes the role of the Spirit, has a potential role to play in minding this gap. The notion of the Earth as Spirit-filled, for example, has profound theological implications for a Christian environmental ethic (Riches and Kirkpatrick-Jung 2023, pp. 129–32). Other Pentecostal theologians have likewise contended that the Pentecost was an event for all creation, a fellowship of all flesh, including non-human creatures (Yong 2010, p. 345). This pneumatic theological perspective is in line with a creation-centered worldview in which the Spirit is in the land and its trees, rivers, and animals, not just in peoples (Pattell-Gray and Trompf 1993, pp. 167–88). It is encouraging to note that there are evangelical Christians who advocate for creation care[2].

The biggest problem in terms of selecting Scripture for preaching about climate or other environmental issues, arguably, may lie in ultra-conservative preaching that takes a fundamentalist approach. Some, if not most, preachers from such traditions deny global warming and climate change. Their denial is based on a creationist and biblically literalist view. They contend that the world was created exactly in the way it was described in Genesis (regardless of the many textual contradictions that are obtained between ancient priestly and non-priestly traditions in Genesis 1–11). It is alarming to note that there are such organizations as Answers in Genesis, Creation Ministries International, and the Discovery Institute that are actively promoting the rejection of the science of evolutionary processes[3]. For Christians who support these organizations, scientists who warn of the climate crisis are the ones who are spreading "fake news" and are even labelled as anti-Christ[4]. John MacArthur is one such conservative preacher who unfortunately attracts millions of Christians in the United States and around the world. Through his sermons on YouTube, for instance, he bluntly dismisses any claims of climate crisis. His preaching is based on the supposed infallibility of the Bible and he labels evolutionary science a hoax[5].

In short, there is a serious challenge linking preaching and the climate crisis. From deciding which Scripture verse to choose to how to interpret that verse, significant challenges face the preacher in sermon preparation in an Earth-bound homiletics approach. The following section is a modest attempt to assist preachers in this endeavor. Through an analysis of two sermons, exploring three interrelated homiletical movements in complementary ways, Earth-bound homiletics aims to proclaim God who is at work mending the broken relationship between humans and the rest of creation.

### 3. Movement One: Engaging Scripture

Addressing the climate crisis is essential to Earth-bound homiletics and is a daunting task. Noting the limitations imposed by the Revised Standard Lectionary and the dangers of the free selection of Scripture, in addition to the alarming reality of biblical literalist and climate crisis deniers, it is hard to see the best way to approach Scripture. To add to this, very limited work has been undertaken to select and interpret Scripture from an ecological perspective. Hence, we are faced with some real obstacles to Earth-bound preaching.

However, Scripture plays a central role in addressing the climate crisis in preaching (Kim-Cragg 2023b). It is not an option not to engage Scripture in Earth-bound preaching. An equally important point is that engaging Scripture is never neutral. Throughout the history of the reception of biblical texts, Scripture has been used both to oppress and to liberate; thus, preachers need to be mindful of its double-edged quality (Kim-Cragg 2022, pp. 498–505).

Interpreting the Bible from an ecological perspective is not a novel idea. Many biblical scholars have written creation-centered commentaries (Neril and Dee 2020, 2021; Habel et al. 2011; Walker-Jones 2009; DeWitt 1994; Habel and Wurst 2000). Bloomsbury's Earth

Bible Commentary series has published ten volumes, with more in the works. There have been laudable scholarly efforts to bring ecology into the foreground as a justice issue (Dickinson 2019; Johnson and Wilkinson 2021; Malcolm 2020).

Encouraged by these efforts, in what follows, I will examine some insights into how to engage Scripture to craft an Earth-bound sermon. The sermon included below (see Appendix A) is one example that showcases how to interpret a text through an ecological lens. It features a non-human entity, the Jordan river, as the main character. The text for this sermon is taken from 2 Kings 5:1–14, the story of the healing of Naaman. In this sermon, the preacher HyeRan Kim-Cragg interprets the text in ways that highlight the river as an actor in the drama of the divine salvation and help the audience to see how rivers themselves can be considered as agents of God's redemption. The sermon was delivered during the opening worship at the Academy of Homiletics Conference on 1 December 2023. The audience comprised members of the guild, most of whom serve as professors of preaching in their theological schools and are actively involved in preaching and research in homiletics.

The sermon begins with a question that helps the audience to recognize the limitations of traditional readings of the passage: "By centering Naaman, what other groups are underrepresented, or overlooked?" Then, the sermon unpacks the text, closely featuring various characters in the story, starting with the slave girl, the prophet Elisha, the wife of Naaman, and his servants. The sermon lifts up their contributions to the healing of Naaman but then moves to the next section with another question, "Did we miss anybody else in this story?" Leaving this next question to the side for the moment, the sermon makes the congregation wait before an answer is suggested, an effective use of a "hook", according to Sondra Willobee (2009, p. 25). Instead, a personal story is interjected, which features a river prominent in the city where the preacher grew up. This personal story connects with the biblical story, and provides a clue to the question, "Did we miss anybody else in this story?" The sermon, however, does not return to the text just yet but invites the congregation to look at the "glocal" (global and local) reality of rivers by addressing the issue of water shortages and the lack of clean water in various places around the world. This enacts Movement Two of Earth-bound homiletics, that is, situating glocal realities which we will examine in the next section. Once the sermon exposes the water problem as a glocal reality, it returns to the text, inviting the congregation to look at the text again in a new light. This hermeneutical movement follows Paul Ricoeur's insights about the movement from the first naiveté to critical reflection and finally to a second naiveté (Allen 1998b, pp. 196–97).

Returning to the text, the preacher explains that the congregation realizes that it was the river Jordan itself that had the power to heal Naaman. The preacher then makes a theological claim that is creation-centered and God-centered:

> Yes, the water knows us as God knows us. Through the water, we sense the awesome mystery of God. We encounter and experience the amazing grace of God. We learn the resilience of life. We appreciate the beauty. We see how it carves a way out of no way, patiently over thousands of years, tunneling through granite. We quench our thirst. We watch things grow. New life! We tap into our wild imagination. We arouse our childlike playfulness and curiosity. Could it be that God used all these aspects of water to heal Naaman? The mystery, the grace, the resilience, the patience, the beauty, the expectation of new life, the sustaining force, the wild childlike imagination, the restoration of flesh like that of a baby?

One of the listeners who heard this sermon presented a paper in the Academy of Homiletics annual conference in the following year. In that paper, Rhody Walker-Lenow offered three homiletical insights and strategies for preaching in a world ravaged by climate change and the politics that surrounds it. She used this sermon as an example of a prophetic sermon which can speak calmly in the midst of a deep crisis, rather than using extreme or alarmist language in its delivery. Walker-Lenow suggests that a sermon dealing with the climate crisis needs a steady tone rather than an alarmist approach that would flounder

on the rocks of political skirmishes. She writes that the preacher wonders aloud what voices have been left out of the story of the healing of Naaman from 2 Kings 5:1–14. She seeks to decenter Naaman in her preaching, and instead wants to highlight the slave girl, whose agency is integral but underrecognized, and the saving action of the water itself. The primary mode of the sermon then, is not assertion or command, but *wonder from the margins*. Surely, confidently, and deftly, the preacher weaves the stories of the girl and the natural world into a story that has been coopted by much louder voices... As preachers navigate the climate crisis, one of their most urgent tasks will be to cut through the noisy voices that would either silence, minimize, omit, or diminish the effects of climate change and the human role in them. Kim-Cragg offers a way forward: in hell, the quiet voice may be heard most clearly (Walker-Lenow 2023).

### 4. Movement Two: Engaging the Situatedness of Glocal Contexts

The second movement of Earth-bound homiletics is to zoom in on the particular context in which the climate crisis is taking place. If Movement One is about engaging Scripture as a way of tapping into the biblical past, Movement Two focuses on exposing a present situation by naming a particular and current "glocal" reality (Robertson 1995, pp. 25–44), whether it be drought, wildfire, deforestation, flood, extreme weather, animal suffering, or extinction, as an example. The term "glocal" is a portmanteau word combining global and local. It sheds light on the fact that there exist sociocultural and economic phenomena in which the distinction between global and local is blurred. Such homiletical engagement of a glocal reality can be performed as an illustration, an analysis, or a testimonial.

The second Earth-bound homiletical movement seeks to understand the current reality in intersectional ways. Creation Justice Ministries, for example, has modelled this kind of intersectional approach with their 2022 webinar entitled "Environmental Justice is Racial Justice: Faith Communities Respond"[6]. Individual oppressive realities do not exist in a vacuum or stand alone. They appear in interlocking ways. One cannot address the climate crisis without also considering the impact of the colonial legacy or the consequences of racism and sexism, not to mention poverty. It is not a coincidence that the poorest places and places where the highest proportions of Indigenous and racialized women and children, as well as the elderly populations, reside are the regions that are most directly and devastatingly impacted by the climate crisis.

To tackle the climate crisis in intersectional ways in preaching means to encourage congregations to see their local reality, including their own lived experiences, in connection with a global reality (White-Hammond 2022). This interconnected optic is critical; seeing the climate crisis in a multifocal way gives congregants tools to understand complex realities and discern the actions of the Holy Spirit within them.

A further perplexing phenomenon needs addressing for Earth-bound preaching, namely, the gap between what preachers preach and what congregations hear when the climate crisis is spoken of. In 2022, the Pew Research Center conducted a survey of 10,156 Americans in the United States to ascertain how religion intersects with Americans' views on the environment[7]. In the same year, another survey was conducted through the pilot program called "EcoPreacher Cohort" gathering responses from nearly 100 preachers and close to 200 lay members who identified these preachers as their leaders in their congregations. A total of 90% of the respondents who were preachers said they had preached at least one sermon about creation addressing the climate crisis in the previous twelve months, but only 31% of the congregation members recalled hearing such a sermon in that same time frame. One might argue that a parishioner may not have heard the sermon if they missed the one time the preacher had addressed the topic. But 61% of these preachers said they addressed climate crisis issues four to six times in the previous twelve months. Congregants, therefore, had more than one opportunity to hear the topic addressed from the pulpit. Yet only a third of them did (Schade et al. 2023). These data support the assertation that there is a gap between what preachers are saying and what congregants are hearing when it comes to the climate crisis and environmental issues. According to

"The Listening to Listeners Project"[8], there is a gap in perception between what preachers think they said and what parishioners heard them say (McClure et al. 2004). This gap may be heightened as regards differences of expectation and understanding when it comes to concern for the environment.

To bridge this gap is a goal of Earth-bound homiletics. In this movement. preachers may have to include more learning opportunities for people. It may not be sufficient to simply preach about global warming from the pulpit on Sunday. Additional and complementary efforts outside the sermon could have a tremendous impact on congregants' ability to hear and respond. For instance, it may be helpful to invite people into sacred spaces where they can talk about their eco-grief, their concerns for their community, and their anxiety about the future of this planet. Offering book studies, hosting community dialogues, and inviting personal reflections as a part of liturgy can help people realize that they are not alone in their concerns and that they have companions on this journey toward climate justice.

At the same time, preachers should equip themselves and their congregations to communicate about the climate crisis effectively and accurately. A program such as ecoAmerica's Blessed Tomorrow Climate Ambassadors offers free, on-demand training and resources to help inform congregations. Their accurate climate science information and biblical and theological language can help inspire action in congregations and communities[9]. Part of this effort includes an intentional effort to counteract the toxicity of the biblical literalist and fundamentalist influences discussed above.

Helping the congregation locate themselves by drawing from their own memories may be another effective homiletical strategy (Mulligan et al. 2005). The sermon shared in the discussion of Movement One above included a personal childhood memory of a river as a testimonial. It was used to evoke empathy and conjure other memories in the congregation. Drawing from their own lived experiences, preachers can help congregations identify their own locations and broaden their awareness and perspectives to see the problem of the climate crisis in terms that are the most real for them. By telling a personal story, preachers can help hearers recall their own stories and make connections with the biblical story that make sense in their own terms.

Walker-Lenow notes the importance of this connection. She observes that the preacher connects her personal story with the biblical narrative, especially the story of the slave girl that has been overlooked "in dominant narratives about Naaman's healing", and as she shares her story, she weaves it into the story of the water with which she has been acquainted in her life—water from the Han River, water in Northern Ontario freshwater lakes, and water in East Africa. Each of these bodies of water has faced pollution, and she gives voice to the eco-violence they have faced. The Preacher Kim-Cragg is aware of the world that would seek to mask these injustices with a narrative proffered by the West that seeks to exonerate themselves. But her voice shoots through the noise and tells a simple, true story. Water is part of God's creation; God uses water to heal and to baptize; God wills the redemption of water (Walker-Lenow 2023).

In short, the Earth-bound homiletics' Movement Two draws our attention to glocal situatedness. This movement helps to identify geographical locations from which most members are listening to a sermon. It also demonstrates how preachers makes an intentional effort to illustrate a "glocal" reality out of their own lived experience in a way that can connect those in the pew with the biblical story and congregations' own lived experiences.

## 5. Movement Three: Engaging Subjectivities (Agency) of the Creation

Movement Three of Earth-bound homiletics is to incorporate Indigenous wisdom, called "lifeways", into preaching that has emerged from Canadian Indigenous wisdom. According to John Grim, "lifeways" are created through action and engagement with Indigenous relations; the notion of lifeways is "an interrogative concept that raises questions about the ways in which diverse indigenous communities celebrate, work towards, and reflect on the wholeness as a people... These reciprocal ways of knowing in indigenous

lifeways manifest difference in expression and underlie the wisdom and the specificity of indigenous knowledge" (Grim 2008, p. 88).

Lifeways as Indigenous Peoples' lived epistemologies are non-binary, cyclical, flexible, and resilient. They are informed by relationships that are cultivated through ceremony and physical interactions with creatures and entities beyond the human family, such as with animals, plants, rocks, rivers, forests, and specific places.

Métis scholar Paul Gareau from Canada asserts that sacredness has less to do with a binary between sacred and profane and more to do with how Indigenous Peoples (both individuals and nations) engage in kinship relations with "more-than-humans", referring to other-than-humans (Gareau 2021, pp. 138–39). This acknowledgement of the agency of more-than-human beings is connected to Indigenous sovereignty, self-determination, and agency. The sovereignty of Indigenous Peoples and their nations include the capacity to resist normative discourses of settler colonialism, whereby the land and more-than-human beings are reduced to natural resources, mere objects to be extracted and exploited for the benefit of a few human people in power. Creaturely capacity and resistance are fully exercised in relation with one another and funded by interdependent and reciprocal relationships through "all my relations" principles, where the agency of more-than-human kin emerges and is honored. This term is originated from the Mohawk phrase "Akwe Nia'Tetewá:neren"—all my relations—and appears on the crest of the United Church of Canada[10]. It signifies and underscores Indigenous peoples' relationality.

Gareau's insights find resonance in the work of Lutheran theologian H. Paul Santmire, who was influenced by Jewish scholar Martin Buber's I-Thou relational theology. Santmire proposes "I-Ens relationships", a notion similar to Indigenous concepts of relationship in that it focuses on the relationality between humans and more-than-humans. Here, "Ens", from the Latin word meaning "being", could encompass the multitude and abundance of creatures, whether they be trees, or rocks, or animals (Santmire 2000, pp. 70–73). From such Indigenous perspectives as Gareau presented here, the vision of Isaiah, "the mountains and the hills before you shall burst into song, and all the trees of the field shall clap their hands" (55:12, NRSVUE), will not be understood merely as a metaphor but as an affirmation of the agency of these entities. Hence, to defile creation is to carry out damage to creation's witness to God as creator (DeWitt 1994, pp. 53–54).

Indigenous lifeways recognize the knowledge of more-than-human beings. Lifeways acknowledge that humans are dependent on more-than-humans; therefore, humans ought not to regard themselves as dominant over more-than-humans but should be humbly open to receiving and learning from them (Linzey 1995). This humility is matched by a desire to live in harmony. Coming from the African continent, Zambian Indigenous scholar Kapya Kaoma seems to share a similar insight that states, "Humanity living in harmony with the environment has always been part of African Indigenous culture and religion... As Africans, we pride ourselves as the daughters and sons of the soil. Therefore, the destruction of the Earth means our own death and ultimately life, as we know it" (Kaoma 2015, p. 3). While it is important to note that Indigenous worldviews are not homogenous but may differ in various regions that carry different histories and lived experiences, one may suggest that Indigenous epistemology honors more-than-human kin as knowers and unsettles the anthropocentric mode of preaching (White 1967). Instead, it encourages the preacher to listen deeply to the voice of the more-than-humans in composing and delivering a sermon. More-than-humans are part of the story of salvation as well, and preachers should fully identify them as kin and neighbors to humans.

Earth-bound homiletic Movement Three engages the subjectivity of the creation in this regard. It not only underscores the symbiotic relationship between humans and more-than-humans but also suggests how this relationship informs and shapes the interrelatedness between the sermonic form, the homiletical process (how the sermon is crafted), and the content of the homily (what is said). In this interdependent and holistic approach to Earth-bound homiletics, we can also recommend the first-person narrative form of preaching where more-than-humans are somehow empowered to speak.

In this regard, Leah Schade's sermon "I Am Water, I Am Waiting" may be useful to showcase as an example of Earth-bound homiletical Movement Three (Schade 2015, pp. 168–87). Scripture passages for her sermon are taken from Genesis 1:1–23; Romans 8:18–25; and John 4:42. Her sermon features water as the narrator and the central actor in the sermon. Schade uses the sermonic form of the monologue, as if Water were telling us, the hearers, the story. This homiletical form is not new but has emerged from the tradition of preaching as testimony. Others have underscored the power of testimony for preaching that flows from its narration of events and first-person confession of belief (Florence 2007, p. xiii). Monologue is also a way of retelling the Bible story in the form of homiletical exegesis (Kim-Cragg and Choi 2013). What is unique in Schade's homiletical attempt is her inhabitation of the voice of a more-than-human kin. This allows Water itself to become the preacher and witness of the Gospel. Water as the narrator speaks of what is seen. Water as the subject proclaims the gospel message as a part of the long procession of bearing witness. In Earth-bound homiletics, where we recognize more-than-humans as agents of belief who can preach, preaching becomes "but one piece of testimony of believers" (Simpson 2008, p. 423).

The first-person narratives that inhabit the voice of more-than-humans resonates with Indigenous epistemologies and "all my relations" principles. This homiletical movement basically assumes that more-than-humans know and they have things to teach humans. Their agency is key to tackling the climate crisis because it decenters the human and challenges the human sense of superiority without letting humans off the hook for their responsibilities vis à vis the creation. What it does is reset the reciprocal interdependency between human and more-than-human communities and call for mutual commitment to overcome this crisis.

Schade is cognizant that in her sermon we hear Water's point of view. Water has something to say and a warning to give. In this sermon, Water has the agency of life and that of death. Water remembers, mourns, gives birth, and finally waits. As with the first sermon discussed above, which also highlighted the agency of water, the Jordan river to be precise, Schade offers a similar message: water carries memories, water knows and does, and it heals again and again.

Finally, in the Earth-bound homiletics of Movement Three, a deepened appreciation of more-than-human kin's subjectivity and their agency is promoted by encouraging humans to be in touch with the creation. Worship beyond the church walls will be a natural homiletical practice leading to worship services that engage with creation in new ways, at times abandoning the church building in favor of worshiping outside, for example (Dahill 2016a, 2016b). Outdoor activities are effective for enhancing Earth-bound preaching.

During the COVID-19 global pandemic, when indoor worship was not permitted, many preachers recorded their sermons outside the church sanctuary, including in the street and close to nature (MacLean and Sunday n.d., 4:35–13:10). Some delivered walking sermons (Kim-Cragg n.d., 10:05–24:24). As online preaching in a hybrid format has become a norm in many churches today, attempts to preach in the forest or in the garden or by the river may be not only possible but desirable. Leah Schade's "Who Is My Neighbor?" Mapping Exercise may be helpful to implement in this endeavor (Schade 2022, pp. 1–12). By preaching outside, we become aware that the hearers of the sermon include more-than-human beings as well as humans. This outdoor homiletic proclamation aims to open the eyes, ears, and hearts of the congregation to animals, trees, all living in the land, and the basic elements of life, including air, fire, and water.

## 6. Conclusions

The author has proposed three movements for Earth-bound homiletics to help preachers and congregations address the climate crisis: engaging Scripture, engaging global and local (glocal) particular contexts, and engaging the subjectivity of the creation by lifting up Indigenous lifeways, worldviews, and epistemologies. Each movement could serve as an introduction, a middle section, or a final section when crafting a sermon. And this homileti-

cal preparation does not have to be a solitary act of the preacher but should be strongly encouraged as a collective exercise that calls for the active participation of the congregation, including tapping into the wisdom and the agency of our more-than-human kin.

**Funding:** This research received no external funding.

**Institutional Review Board Statement:** This research did not require institutional review.

**Informed Consent Statement:** This research did not require informed consent.

**Data Availability Statement:** There are no additional data available for this research.

**Conflicts of Interest:** The author declares no conflicts of interest.

## Appendix A

**Sermon Title: Water Does It Again: The Healing of Naaman (2 Kings 5:1–14)**
Preached by HyeRan Kim-Cragg on 1 December 2023 at the Academy of Homiletics Conference in Louisville, Kentucky, USA

Let us pray: Dear God incarnate, open our hearts and minds with humility so that we can hear your healing message.

Some headlines in the news can mislead. They fail to capture the whole story.

The headline of the story we heard today is "the healing of Naaman". By centering Naaman, what other groups are underrepresented, or overlooked? Yes, it is true that the biblical story is about Naaman, but we know there are so many others who are involved in this tale as well.

The journey of Naaman's healing actually started with a young girl captive from the land of Israel. Nothing would have happened if this slave girl had kept her mouth shut. It was in her bold utterance that everything was made possible. She said to her mistress, "If only my lord were with the prophet who is in Samaria! He would cure him of his skin disease".

This unnamed slave girl had the courage to speak. This war captive had the grace to care for her master. Her country had lost a war against the Syrians and that is how she became a slave in a country of Israel's oppressors. This young girl is serving the wife of the man who probably orchestrated the killing of so many of her people. Many of them would have been innocent of any violence. This girl would have watched the destruction of her city. If I were her, I would have been unwilling to do anything for this man's illness. Yet, she wanted to help in the work of healing.

Where does this bold courage and amazing grace come from? Would it come from her faith in God? "If only my lord were with the prophet", she uttered. So the headline of this story could be,

"*Slave Girl's Courageous Evangelism*" or

"*Victim of Violence Loves Her Enemies*" or

"*Utterance of Grace Heals*". We could reimagine the story featuring this slave girl as the central character.

Let's also not forget the humility of the wife of Naaman, who listened to her maid. As madam, she could have easily dismissed her. This wife could have even disciplined her maid for her boldness, but she didn't and instead accepted her advice and acted upon it. That takes humility, so the headline of this story could be about humility, a lesson that those of us in the center should heed. The headline could be then something like,

"*Powerful Woman's Humility Leads to Renewed Health.*"

One more group that has not been centered in our homiletical imagination is the servants of Naaman. Naaman was angered by the prophet Elisha's lukewarm reception. He was furious and ready to go back home. In fact, he was in the process of turning away from Elisha's house. In that critical moment, his servants stopped him from going further. They did this with great persistence. So the headline of this story might be,

"*Servants Save the Day*" or

"*Love of the Lowly Corrects the Mighty*". In many remarkable historical events, it is often those whose names never appear in the history book as heroes, yet we know they were instrumental in making the events possible.

When I served the Women in Leadership in the Association of Theological Schools (ATS), I heard so many stories about how women saved their respective institutions yet often never were given credit for what they had done, let alone bestowed with public honour for their hard and courageous work.

This, I believe, is because theological schools are still not ready to appoint women as their president, principal, or dean. I know it is not just women but also those who belong to minorities.

Today, therefore, I would like to acknowledge and lift up those forebears and trailblazers who paved the way for some of us as women, racialized, and queer members who are called to serve as the head of their schools. Many women racialized and queer leaders have been cultivated in this organization, our Academy of Homiletics.

Yes, it is true that the story of 2 Kings chapter 5 is about the healing of Naaman. And it is also the story about the slave girl, the prophet Elisha, the wife of Naaman, and his servants, all of whom played a part. They all made a difference in the wellbeing of this sick man, Naaman. But who else? Did we miss anybody else in this story?

I was born and grew up in South Korea, to exact, mainly two places, Seoul and Inchon.

20% of the population lives in Seoul, a tiny land, in an area three quarters the size of Louisville, Kentucky. Seoul is one of the most densely populated cities in the world. But the main feature of the city is the Han River.

I was so happy to go back to Seoul this past summer and had a chance to walk along the river. Every time I was on the subway that crossed the river, I realized how peaceful and inviting this River is to all who need healing. I was brought back to the past, recalling the memory of my ancestors. I was reminded what this river does and knows.

In times past, it was a river for trade. It was a river that enabled scholars to travel. They were looking for rest from the efforts of urban living in the capital and sought peace and solitude in the beautiful green mountains of the interior.

It was also a river that witnessed the unwelcome penetration of Western, U.S. and French imperial forces. It was the river who knew how their gunboats imposed unjust treaties upon the Korean people in the 18th and 19th centuries.

It was this frozen river across which four million war refugees crossed in the winter of 1951, in the depths of the Korean War. It was this River that ushered my parents and grandparents who were part of that great upheaval into a safer place. For them, this was a new home in the south.

For others who had espoused socialist ideals or opposed the U.S.- and Japan- friendly government in the south, it was the path to the north to a haven for them from persecution and unspeakable violence.

Today, it is the river where millions of people come out to rest and walk amidst the busy work daily.

It is the river that inspires imagination and sparks ideas for those of us who feel stuck in the middle of preparing sermons, writing journal articles and monographs, marking term papers, reading dissertations, and so on.

It is the river that calms our troubled souls when we are in conflict, struggling with broken relationships and seeking healing.

It is the river where birds, fish and other creatures find shelter and food.

It is the river that provides drinkable water, the most important necessity of life to all who need it, people and other creatures.

Korea is surrounded by the Pacific Ocean, but this abundant ocean water cannot help quench thirst. Unlike Canada and the U.S., blessed with lakes, Korean people and all living on this land would not survive without the fresh water coming from the rivers.

One of the most astonishing stories I heard on the news is that many Indigenous people in Canada are getting sick because they don't have clean water to drink. Canada is

the wealthiest country in terms of having the four biggest freshwater lakes. Yet some of the first people whose ancestors lived on this land for thousands of years are not able to access clean water.

Equally disturbing news about the lack of clean water comes from Africa. Many people there, especially East Africa, know that the water they have is toxic and contains bacteria, including E. coli. Yet they still drink it and get sick and die. Severe drought was created by human greed, the ravaging force of industrialism, the truth-denying and science-concealing greed of neoliberal transnational capitalism. These evil systemic forces are drying up wells and rivers in that region. In Somalia, one out of every five children faces death from hunger caused by drought, news reports have recently revealed.

In Europe, drought is uncovering hunger stones in the beds of rivers that have not been exposed for centuries. People who experienced the hunger caused by drought left records of their suffering on these stones before these stones were mercifully submerged once again in the waters of life. Due to the severe drought, the stones at the bottom of these rivers are once again appearing and crying out, calling us to stop this greed and asking us to challenge the deeply flawed system.

Is water becoming a commodity? Is water being used as a weapon of colonialism and racism?

These questions beckon us back to the biblical story for today. Let me ask you again, circling back to the question, "did we miss anyone else in the story?"

"What made the healing of Naaman actually possible? It was the water at Jordan."

Patty Krawec, the Indigenous author of a book entitled, *Becoming Kin: An Indigenous Call to Unforgetting the Past and Reimagining Our Future*, shares her personal story about going back to her birthplace called "Sioux Lookout", a Northern Ontario indigenous place in an interview of the CBC Radio program called "Tapestry"[11].

> It was the first time I had been home since leaving as a toddler, and I did not know what to expect. I spent a lot of time going down to the water... And I remember still kind of leaning down and putting my hands in the water and it was like a physical reaction that I hadn't anticipated.... I felt remembered, like I wasn't the only one who was remembering. And I felt the water knew me. And I think about the way the waters from Sioux Lookout go down to Lake Superior, and then through the Great Lakes and pass by where I am now. And then the water cycles back in the form of rain. And it makes me think of this great conversation that's going on between the land and the water about us.

> Yes, the water knows us as God knows us.

> Through the water,

> we sense the awesome mystery of God.

> We encounter and experience the amazing grace of God.

> We learn the resilience of life.

> We appreciate the beauty.

> We see how it carves a way out of no way, patiently over thousands of years,

> tunneling through granite.

> We quench our thirst.

> We watch things grow. New life!

> We tap into our wild imagination. We arouse our childlike playfulness and curiosity.

> Could it be that God used all these aspects of water to heal Naaman?

> The mystery,

> the grace,

> the resilience,

> the patience,
>
> the beauty,
>
> the expectation of new life,
>
> the sustaining force,
>
> the wild childlike imagination,
>
> the restoration of flesh like that of a baby?
>
> And are all of these not aspects of our healing, too?
>
> The headline for this story in the Bible is "the healing of Naaman."
>
> And maybe it should be "Water does it again."
>
> Thanks be to God.

## Notes

1. The Season of Creation is one such commentary (https://seasonofcreation.org/) accessed on 15 December 2023. Lutherans Restoring Creation has a section on their website with Creation-based lectionary commentaries: https://lutheransrestoringcreation.org/worship/lectionary-commentaries/ accessed on 16 December 2023. There is also the Green Lectionary podcast: https://www.creationjustice.org/green-lectionary-podcast.html accessed on 16 December 2023. The Interfaith Center for Sustainable Development offers the EcoPreacher 1-2-3 resource for lectionary texts: https://interfaithsustain.com/ecopreacher-123/ accessed on 19 Decemeber 2023. The Church of England has a Greening the Lectionary project: https://www.greeningthelectionary.net/ accessed on 19 December 2023. The Anglican Diocese of New Westminster's project, Salal and Ceder, includes a Wild Lectionary resource: https://www.salalandcedar.com/wildlectionary accessed on 20 December 2023. The Anglican Communion Environmental Network offers Preaching for God's World with reflections on the lectionary readings from the perspective of ecological justice: https://preachingforgodsworld.org/ accessed on 20 December 2023.

2. Katharine Hayhoe is a leading climate scientist who is also an evangelical Christian and has dedicated her work to bridging the gulf between faith and climate science. Two of her books are *A Climate for Change: Global Warming Facts for Faith-Based Decisions* (Hayhoe 2011) and *Saving Us: A Climate Scientist's Case for Hope and Healing in a Divided World* (Hayhoe 2021). Matthew Sleeth has written several books, including *Reforesting Faith: What Trees Teach Us About the Nature of God and His Love for Us* (Sleeth 2019), and *Serving God, Saving the Planet: A Call to Care for Creation and Your Soul* (Sleeth 2007). There is also an organization called the Evangelical Environmental Network that advocates for creation care: https://creationcare.org/ accessed on 20 December 2023. And there is the Young Evangelicals for Climate Action: https://yecaction.org/.

3. http://answersingenesis.org; http://creation.com; http://www.discovery.org accessed on 21 December 2023.

4. https://theconversation.com/god-intended-it-as-a-disposable-planet-meet-the-us-pastor-preaching-climate-change-denial-147712 accessed on 21 December 2023.

5. https://youtu.be/ZTlYl8E_B14; https://www.youtube.com/watch?v=XVq8fYdiwCQ&t=49s accessed on 22 December 2023.

6. https://www.youtube.com/watch?v=XBdCXypEiYo accessed on 22 Decemeber 2023.

7. Pew Research Center, November 2022, "How Religion Intersects With Americans' Views on the Environment". All respondents to the survey were part of Pew Research Center's American Trends Panel (ATP), an online survey panel that is recruited through the national random sampling of residential addresses. The survey is weighted to be representative of the U.S. adult population by gender, race, ethnicity, partisan affiliation, education, religious affiliation, and other categories.

8. The Listening to Listeners Project, a Lilly Foundation-funded research project led by Ronald J. Allen from 2001–2002, focused on how parishioners listen to sermons. This was a time-honored project, in my view, as listening to listeners is critical for preachers and parishioners navigating this unprecedented time of addressing the climate crisis.

9. Blessed Tomorrow is a coalition of diverse religious partners working to advance climate solutions in faithful service to God. It is a program of ecoAmerica, a network of major institutions and thought leaders in five sectors—faith, health, communities, higher education, and business—who have the power to inspire tens of millions of Americans on climate change, in counties and communities nationwide. https://blessedtomorrow.org/blessed-tomorrow-ambassadors-training/ accessed on 22 Decemeber 2023.

10. https://united-church.ca/community-and-faith/being-community/indigenous-ministries/all-my-relations accessed on 22 Decemeber 2023.

11. Krawec (2022) https://www.cbc.ca/listen/live-radio/1-59-tapestry/clip/15947963-becoming-kin-indigenous-author-patty-krawec accessed on 15 October 2023.

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
