# Peer review of "Earth-Bound Preaching: Engaging Scripture, Context, and Indigenous Wisdom"

_religions, doi:10.3390/rel15030357_

Round 1

Reviewer 1 Report

Comments and Suggestions for Authors

This is a much-needed contribution to the field of homiletical theory and praxis.  The author makes the case for why “earth-bound preaching” is necessary and then lays out three “movements” for crafting sermons that address the climate and environmental crises of our time.  This will be a helpful article for preaching professors to share with their students when assigning them to preach an environmental sermon.   Including the two sermon examples is a good way to illustrate the kind of “earth-bound preaching” the author has in mind.

General:

1.       Could the title of the piece include a subtitle so that readers can get a better sense of what the article is about?  Maybe: Engaging Scripture, Context, and Indigenous Wisdom

2.       Sometimes the term "creation" is capitalized, other times it's not.  I recommend capitalizing, but consistency would be good either way.

Line-specific:

1.       48-49: It would be good to list in an endnote the works in homiletical literature that HAVE dealt with the climate crisis and other environmental issues.  Here’s a suggested list:

a.       Dieter Hessel, For Creation's Sake:  Preaching, Ecology and Justice (Philadelphia: Geneva Press, 1985).

b.       Elizabeth Achtemeier, Nature, God, and Pulpit (Grand Rapids, Mich.: W.E. Eerdmans Pub. Co., 1992).

c.       Stan L. LeQuire, editor, The Best Preaching on Earth:  Sermons on Caring for Creation (Valley Forge, PA: Judson Press, 1996).

d.       David Rhoads, editor, Earth and Word (New York: Continuum, 2007).

e.       Norman C. Habel, David M. Rhoads, H. Paul Santmire, editors, The Season of Creation: A Preaching Commentary (Minneapolis, MN: Fortress Press, 2011).

f.        John C. Holbert, Preaching Creation: The Environment and the Pulpit (Eugene, OR: Wipf & Stock, 2012).

g.       Leah D. Schade, Creation-Crisis Preaching: Ecology, Theology, and the Pulpit (St. Louis, MO: Chalice Press, 2016).

2.       68-70: I'm not sure what you mean by this sentence or what you’re referring to.  What do you mean by “connected to broader political commitments”?  What does it mean that preaching on the climate crisis “must be dealt with structurally”?  Can you elaborate briefly?

3.       71: after the word “preaching” add: about climate or other environmental issues.

4.       83-85: Do you want to say something about the Season of Creation three-year lectionary cycle?  (See note above regarding the preaching commentary.  Also, they have a website:  https://seasonofcreation.org/.)  And acknowledge the efforts of some to "green the lectionary," (i.e., Lutherans Restoring Creation has a section on their website with Creation-based lectionary commentaries: https://lutheransrestoringcreation.org/worship/lectionary-commentaries/. There’s also the Green Lectionary podcast:  https://www.creationjustice.org/green-lectionary-podcast.html. The Interfaith Center for Sustainable Development offers the EcoPreacher 1-2-3 resource for lectionary texts: https://interfaithsustain.com/ecopreacher-123/. The Church of England has a Greening the Lectionary project: https://www.greeningthelectionary.net/. The Anglican Diocese of New Westminster’s project, Salal and Ceder, includes a Wild Lectionary resource: https://www.salalandcedar.com/wildlectionary. The Anglican Communion Environmental Network offers Preaching for God’s World with reflections on the lectionary readings from the perspective of ecological justice: https://preachingforgodsworld.org/.)

5.       96-97. The sentence should read:  “The status quo may include the failure to address the climate crisis.”

6.       106: “minding”? Do you mean “mending”?

7.       114: after the word “preaching” add: about climate or other environmental issues.

8.       115: It's important to note (at least in an endnote) that there are, in fact, evangelical Christians who advocate for Creation care. For example, Katharine Hayhoe is a leading climate scientist who is also an evangelical Christian and has dedicated her work to bridging the gulf between faith and climate science. Two of her books are A Climate for Change: Global Warming Facts for Faith-Based Decisions (Faithwords, 2011) and Saving Us: A Climate Scientist's Case for Hope and Healing in a Divided World (Simon & Shuster, 2021). Matthew Sleeth has written several books, including Reforesting Faith: What Trees Teach Us About the Nature of God and His Love for Us (New York: WaterBrook, 2019), and Serving God, Saving the Planet: A Call to Care for Creation and Your Soul (Zondervan, 2007).  There’s also an organization called the Evangelical Environmental Network that advocates for Creation care: https://creationcare.org/. And there’s the Young Evangelicals for Climate Action:  https://yecaction.org/.

9.       129-130: The sentence should read:  From deciding which Scripture verse to choose . . .

10.   140-141: The sentence should read: Add to this the very limited work that has been done. . .

11.   162: “help” should be “helps”

12.   187: The sentence should read: Returning to the text, the preacher explains that . . .

13.   157-188: It’s odd that the author is referring to themself in the third person when writing about their own sermon.  I think it would be better to claim this as a sermon as their own and use first-person language.  Perhaps that will be fixed when the non-anonymized version is finalized.

14.   217: There needs to be a citation for Walker-Lenow’s paper.

15.   257: Suggest the sentence should read:  This data supports the assertion that . . .

16.   299: There needs to be a citation for Walker-Lenow’s paper.

17.   308-314: It would be good to include the term “all my relations” in this paragraph since this is a key part of the author’s argument.  Also, maybe I missed it, but it’s not clear to me where this term originated.  Who first coined it and when?  What does it mean?  Could the author provide a concise definition or description in this paragraph?

18.   314: Seems to be an incomplete citation.  Shouldn’t the author’s last name be included so that readers can trace the reference to the endnote?

19.   336: Seems to be an incomplete citation.  Shouldn’t the author’s last name be included so that readers can trace the reference to the endnote?

20.   379: does “all my relations” need to be in quotations?

21.   Endnote 4: The EcoPreacher 1-2-3 resource does not include references to Islamic texts.  So it would not be appropriate to list this resource as an example of one that incorporates Christian, Jewish, and Islamic texts.

22.   Endnote 7: The second sentence appears to be a quote from something because the “we” doesn’t seem to refer to the author’s work with any group.  If this is a quote, it needs to be noted as such and properly cited. 

Comments on the Quality of English Language

See comments above with edits.

Author Response

I thank this reviewer for a very close reading and helpful suggestions and comments.

I followed everything that was noted by the reviewer either accepting the suggested wording, including a sentence, or adding end notes, except for the following:

 * minding is a correct word

* two citations without the last name are correct because the main texts mention the authors.

Reviewer 2 Report

Comments and Suggestions for Authors

The author provides a very insightful exploration of both theory and practice focused on Earth-bound homiletics. Particularly intriguing were the principles advocating for an eco-centric reading of the Bible based on dehumanization, the exposure of specific local and global ecological realities, and the incorporation of indigenous worldviews and epistemologies into sermon content and form. The author's inclusion of sermon samples applying these principles effectively balanced theory with practical application.

However, I have one request: the author seems to overly generalize indigenous peoples. When an example of African indigenous peoples is mentioned midway through the text, I found myself wondering whether the indigenous people referenced initially were from Africa, Asia, North America, or South America. Not all indigenous peoples across the globe necessarily share the same epistemologies and worldviews described by the author. Therefore, it would be beneficial for the author to provide a more nuanced explanation of which indigenous group(s) the author is referring to and to narrow down the scope accordingly.

Additionally, the author states, "first, the author presents some reasons for why the climate crisis has not been a major topic in Christian theological disciplines and is rarely preached upon in most congregations of any denomination." However, the transition to the section titled "2. Problem of Choosing Scripture for Earth-bound Preaching" feels abrupt and disrupts the flow of the text. The content from lines 45 to 70 seems unrelated to the subsequent section. Thus, reorganizing this portion under a different heading would improve the coherence and structure of the essay. Overall, these adjustments would enhance the clarity and effectiveness of the piece.

Author Response

I thank you for the positive review, especially a note on clarifying particularities of Indigenous groups and their views. I incorporated that, please see (322, 333, 356)

Regarding the section one on problem of choosing scripture,

I laid out clearly in the introduction (please see 28-32). 

Round 2

Reviewer 1 Report

Comments and Suggestions for Authors

Excellent piece!  This will make a positive and important contribution to the homiletics guild and to the task of preaching and environmental issues.